# Plasma 25-Hydroxyvitamin D and 1,25-Dihydroxyvitamin D Levels in Breast Cancer Risk in Mali: A Case–Control Study

**DOI:** 10.3390/diagnostics13243664

**Published:** 2023-12-14

**Authors:** Aboubacar D. T. Bissan, Madani Ly, Awo Emmanuela H. Amegonou, Fatoumata M. Sidibe, Bocary S. Koné, Nènè Oumou K. Barry, Madiné Tall, Lassana G. Timbiné, Bourèma Kouriba, Pascal Reynier, Zahra Ouzzif

**Affiliations:** 1Biochemistry, Metabolic and Molecular Unit, Faculty of Medicine and Pharmacy of Rabat, Mohammed V University of Rabat, Rabat 10100, Morocco; zahra_ouzzif@yahoo.com; 2Charles-Merieux Center for Infectiology (CMIC) of Bamako, Bamako BPE2283, Mali; madine.tall@cicm-mali.org (M.T.); ltimbin67@gmail.com (L.G.T.); bourema.kouriba@cicm-mali.org (B.K.); 3Biology Teaching and Research Department, Faculty of Pharmacy, University of Sciences, Techniques and Technologies of Bamako (USTTB), Bamako BPE423, Mali; awohilda@gmail.com (A.E.H.A.); fatsi_2@hotmail.com (F.M.S.); bocarysidi45kone@yahoo.fr (B.S.K.); 4University Hospital of Luxembourg, Bamako BPE91094, Mali; madanily2003@yahoo.fr; 5University Hospital of Point G of Bamako, Bamako BPE91093, Mali; 6Pharmaceutical Biochemistry Laboratory, Cheikh Anta Diop University of Dakar, Dakar BP 5005, Senegal; oumou.barry22@yahoo.com; 7Department of Biochemistry and Molecular Biology, University Hospital of Angers, 49933 Angers, France; pareynier@chu-angers.fr

**Keywords:** breast cancer, 25-hydroxyvitamin D, 1,25-dihydroxyvitamin D, hypovitaminosis D, metastasis

## Abstract

(1) Background: Breast cancer is the most prevalent cancer found in women in Mali. The aim of the current study was to determine the association between metabolites circulating in the blood, 25(OH)D and 1,25(OH)_2_D, and vitamin D levels with the risk of breast cancer in Malian women. (2) Methods: We conducted a prospective case–control study from August 2021 to March 2022. Control subjects were matched to cases according to age (within 5 years). The patients’ clinical stage was determined by the oncologist according to the tumour–nodes–metastasis (TNM) classification system. (3) Results: We observed no differences in the mean 25(OH)D (*p* = 0.221) and 1,25(OH)_2_D (*p* = 0.285) between cases and controls. However, our findings indicate a more pronounced inverse association in the first level of plasma 25(OH)D, while the risk function decreases at higher levels. This observation takes strength with 1,25(OH)_2_D by a significant association between the first quartile and breast cancer as a risk factor (*p* = 0.03; OR = 71.84; CI: 1.36–3785.34). (4) Conclusions: These outcomes showed a possible association between 25(OH)D and 1,25(OH)_2_D in decreasing the risk of breast cancer.

## 1. Introduction

Breast cancer (BC) is the most prevalent cause of death by cancer among women [1], particularly in Africa [2]. In Mali, breast cancer represented 24.8% of cancers in women in 2020 according to the GLOBOCAN analysis report, making it the most common form of cancer in women in the country [3].

For a long time, vitamin D (vitD) has been understood to have certain effects on phosphocalcic metabolism and bone mineralisation. Our fundamental and clinical knowledge of the multitissue influence of this steroid has, however, evolved in recent years. The natural form of vitamin D, vitamin D_3_ (cholecalciferol), is obtained from dietary sources and is also generated in the human skin under the influence of sunlight (ultraviolet B radiation) from its precursor, 7-dehydrocholesterol [4,5,6]. Two hydroxylation steps generate the biologically active form of vitamin D_3_, the 1α,25-dihydroxyvitamin D (1,25(OH)_2_D). The first hydroxylation starts with carbon 25 by CYP2R1/CYP27A1 (cytochrome P450 enzymes) in the liver and the next in the kidney by CYP27B1, which hydroxylates 25-hydroxyvitamin D (25(OH)D) at carbon 1. 1,25(OH)_2_D_3_ exercises its biological functions by binding to the vitamin D receptor (VDR), a transcription factor that regulates gene expression in vitamin D target tissues [7,8]. It is known to have multiple antiproliferative, proapoptotic, prodifferentiating, and anti-inflammatory effects on malignant cells, as in BC [7].

However, epidemiological evidence for the relationship between plasma 25(OH)D and BC incidence is limited and conflicting [9,10]. Indeed, several longitudinal studies on serum 25(OH)D and multiple cancer risks have concluded that 25(OH)D concentrations are inversely associated with the incidence of colorectal cancer [10,11] but not with prostate cancer or BC incidence [10,12,13], while many studies suggest that vitamin D may reduce the risk of BC [14,15,16] and is also subtype-specific [17].

Racial differences in both BC and vitamin D metabolism have been described [17,18]. Dark skin pigmentation may play an important role in the deficit of circulating vitamin D levels [19]. And geography likely plays a part too. Mali is a sub-Saharan country that receives large amounts of sunlight and has prolonged periods of high temperatures (the maximum temperature varies between 34 °C and 37 °C). Its climate is both dry, with the Sahara desert in the north, and tropical, with the Sahel region running through the centre. Some studies have been published on 25(OH)D [15,20] in Africa. To our knowledge, no published studies explore 25(OH)D and 1,25(OH)_2_D levels and their relationship to breast cancer in sub-Saharan countries, such as Mali.

The aim of the current study was to determine the association between the blood-circulating metabolite 25(OH)D and the active metabolite 1,25(OH)_2_D of vitamin D levels with the risk of breast cancer amongst Malian women.

## 2. Materials and Methods

### 2.1. Study Population

This is a prospective case–control study in a population of Malian women. This study was carried out over a period of 8 months, from August 2021 to March 2022. The questionnaire and methodology for this study was approved by the Ethics Committee of the UNIVERSITY OF SCIENCE, TECHNIQUES AND TECHNOLOGIES OF BAMAKO (Ethics approval number: 2021/236/USTTB).

The recruitment of breast cancer cases was carried out after histological confirmation in the medical oncology departments of two university hospitals: the Luxembourg “Mère-Enfant” and that of Point G. Analysis took place at the Rodolphe Merieux Laboratory (RML) at the Charles Merieux Center for Infectiology (CMCI) in Mali.

This current study included 110 women with breast cancer and 110 women without breast cancer for 25(OH)D assays. Randomly, we measured 1,25(OH)_2_D in 35 women from the BC group and 35 women in the control group, with no selection criteria. This study was restricted to females aged 18 and above.

We included all patients living in Mali for more than a year before inclusion with newly diagnosed breast cancer in the 2 departments, regardless of the grade, who had not initiated chemotherapy treatment. The patients’ clinical stage was determined by the oncologist according to the tumour–nodes–metastasis (TNM) classification system [21]. This categorisation divides breast cancer into 4 stages, from I to IV, based on the following criteria: localised invasive breast cancer (stages I and II); inoperable locally advanced invasive breast cancer (stage III); and metastatic disease (stage IV).

Control subjects were matched to cases according to age (within 5 years). They were all apparently healthy, cancer-free women accompanying the patients, coming from gynaecology services or coming to the LRM for their assessment. We included only the women who had evidence of a recent consultation or a follow-up with a gynaecologist attesting to their health or women with normal mammography data. Patients with a history of cancers other than breast cancer were excluded, as were patients with diseases that would prevent them from exposing themselves to the sun. Pregnant, lactating women and women using vitamin D supplements were excluded at the time of enrolment from both cases and controls. Patients whose blood samples underwent haemolysis were also excluded. All patients with and without cancer provided written informed consent, following which they were given a questionnaire to fill in during an interview. All participants were informed about the aim of the study by a general information document.

The questionnaire contained variables regarding information on sociodemographic data, cancer history, parity, chronic diseases (such as diabetes mellitus, high blood pressure, etc.), TNM classification of cases, hormone receptor status (if available), amount and frequency of food consumption, sun exposure status, and physical activity.

### 2.2. Samples Collection

In view of the COVID-19 pandemic, barrier measures were scrupulously respected. A blood sample of 10 mL was collected into 2 sodium heparinate tubes from patients who had fasted for 8 h. The samples collected were placed in a refrigerated bag with a gel pack (allowing storage between 2 °C and 8 °C) and sent immediately to the CMIC. They were centrifuged at 2314 rcf (Relative Centrifugal Force) for 10 min and then aliquoted with at least 200 μL of plasma into cryovials and stored at −80 °C for 7 months until the end of sampling. The samples were anonymised using codes that combined their origin and their order of sampling.

### 2.3. Analysis of Vitamin D Metabolites

The concentration of vitamin D metabolites in the blood was determined by an automatic chemiluminescence method produced on an Immuno Diagnostic System (IDS-iSYS) device (Auxois, France). For the 25(OH)D assay, we used the reagents IDS 25VIT Ds, IDS 25VIT Ds Control Set for control kit and IDS 25VIT Ds Cal Set for calibrators. For the 1,25(OH)_2_D assay, we used the following kits: IDS 1,25VitD Xp, in which calibrators were included, and IDS 1,25VitD Xp Control Set for control reagent.

The 25(OH)D needs a pretreatment step to denature the Vitamin D Binding Protein (VDBP). After neutralisation in a buffer solution, an anti-25(OH)D antibody labelled with biotin was added. After 2 incubation steps, 25(OH)D labelled with acridinium and the magnetic particles bound to streptavidin were added successively. After the last incubation step, the complex was recovered using a magnet and washing was performed to remove any unbound analyte. Activation reagents were added, and the resulting light emitted by acridinium labelling was inversely proportional to the concentration of 25(OH)D present in the starting sample. The interpretation of vitamin D adopted is presented below: deficient level < 20 ng/mL; insufficient level between 20 ng/mL and 30 ng/mL; and normal levels between 30 and 50 ng/mL.

The 1,25(OH)_2_D needs an immunoextraction step. First, in a cuvette, the sample was delipidated and then incubated using magnetic particles coated with specific anti-1,25(OH)_2_D antibodies. After incubation, the magnetic particles were washed and the 1,25(OH)_2_D of the sample was eluted. This eluate was transferred into a second cuvette in which the assay was based on the same chemiluminescent method as 25(OH)D.

The interpretation of 1,25(OH)_2_D adopted is the following normal range: 15.2 to 90.1 pg/mL.

### 2.4. Statistical Analysis

Data were recorded in Excel 2022 and analysed using the IBM Statistical Package of Social Science (SPSS.21) software (IBM corp, New York, NY, USA).

Quantitative variables were expressed as mean ± standard deviation, and qualitative variables were given as number (*n*) and percentage (%). Student’s *t* test was used to compare means, Chi2 test to compare qualitative variables, and Pearson’s r was used for correlation tests. We then used multivariable conditional logistic regression to verify the effects between the qualitative dependent variable and the other factors. The results are considered statistically significant for a value of *p* < 0.05.

## 3. Results

We included a total of 220 patients, consisting of 110 cases of breast cancer that were matched to 110 controls according to age (within 5 years). We excluded 12 patients (6 cases and 6 controls) due to technical issues, so our study population included 208 patients with 104 cases and 104 matched healthy women.

### 3.1. Sociodemographic and Clinico-Pathological Characteristics of the Study Population

The characteristics of the study population are shown in Table 1. The mean age of the study population was 48.09 ± 12.42 years for the cases, ranging from 21 to 85 years, and 47.79 ± 12.42 years for controls, with 20 to 84 years for the range.

The majority of both case and control populations were from urban areas, while 35.6% of the cases and only 4.8% of the controls were from rural locations. There was no difference in the mean age at menarche between the two groups (*p* = 0.715), while we noticed some significant differences in the age at which subjects had their first pregnancy (*p* = 0.001), BMI (*p* = 0.024), parity (*p* < 10^−3^) and in the months of breastfeeding (*p* < 10^−3^). With half of each group consisting of menopausal women, our population exposure time was significantly higher in the BC group than in the control group (*p* < 10^−3^). The majority of the women in the control group had a moderate socioeconomic status (57.70%), while women in the BC group were largely of low socioeconomic status (70.20%) (*p* < 10^−3^).

Moreover, there was a significant difference between the control group and the cases group with regard to a family history of cancer (*p* = 0.04).

The majority of women in our cases group had metastasis (73.40%), including a high number of advanced breast cancer, of which 81.90% were either stage III or stage IV.

### 3.2. Vitamin D Metabolite Levels in Breast Cancer Patients

Across all 208 women, the mean values of 25(OH)D and 1,25(OH)_2_D were similar between the case and control groups (*p* = 0.221 and *p* = 0.285, respectively), as shown in Figure 1. Despite the insufficiency of 25(OH)D in the two groups, 1,25(OH)_2_D remained within the normal range (Figure 1).

There was no correlation between levels of 25(OH)D and 1,25(OH)_2_D among subjects in the BC group, as shown in the scatterplot of Figure 2. This absence of correlation between precursor and active substance suggests the possible presence of unknown mechanisms that could cause their fluctuation in BC women, while the presence of a correlation indicates an interdependence of the levels of the two markers.

Using the 30 ng/mL cut-off of 25(OH)D, the majority of the subjects in both groups were in hypovitaminosis D with 93.9% and 95.2% for cases and controls, respectively. The opposite is found with 1,25(OH)_2_D concentrations for cases and control groups, both of which were largely in the normal range (Table 2).

Among patients with breast cancer, we noticed no difference in 25(OH)D and 1,25(OH)_2_D plasma concentration between subjects with early (clinical stages I and II) and advanced (clinical stages III and IV) disease. Table 3 shows a nonsignificant decrease in these two metabolites from subjects without metastasis to ones with metastasis.

### 3.3. Vitamin D Metabolites and Breast Cancer Risk

In a univariate statistical analysis of our population, we found a number of factors that were associated with breast cancer, as shown in Table 1. We also performed a multiple binary logistic regression model for breast cancer for 25(OH)D and 1,25(OH)_2_D, both stratified by quartiles, as shown in Table 4.

There was no association between metabolites of vitamin D, 25(OH)D, and 1,25(OH)_2_D and breast cancer. However, it is noteworthy that our findings indicate a more pronounced inverse association in the first level of plasma 25(OH)D, while the risk function decreases at higher levels. This observation takes strengths with 1,25(OH)_2_D by a significant association between the first quartile and breast cancer as a risk factor (*p* = 0.03; OR= 71.84; CI: 1.36–3785.34). The risk function flattens and remains the same with increased concentrations.

## 4. Discussion

As the widely accepted marker for its stability and reliability, we chose 25(OH)D as the main metabolite measured in the current study. The second one, 1,25(OH)_2_D, as the biologically active form of vitamin D, was measured to allow us to explain some of the variations in 25(OH)D levels. In fact, the lack of correlation between these two metabolites could make us hypothesize the presence of another unknown mechanism, which could constitute new lines of research.

Several studies have addressed the biological effects of vitamin D metabolites, including 1,25-dihydroxyvitamine D (1,25(OH)_2_D) and its precursor 25(OH)D, on breast cancer [7,22,23], seeking to determine an association with breast cancer disease in diverse populations.

We observed no difference in the mean 25(OH)D (*p* = 0.221) between cases and controls, thus suggesting a lack of association between vitamin D and breast cancer risk. The same results are found in studies by Lyra et al. [24], Janowsky et al. [18], and Eliassen et al. [25], which reported no significant changes in 25(OH)D concentrations. These findings were in contrast to many studies, such as those by Husain et al. [15], Patel et al. [16], Abboud et al. [26], and Khedr et al. [27], which detected a significantly lower concentration of 25(OH)D in subjects with breast cancer. These differences could be explained by the differences between the populations included in these studies, especially with respect to skin pigmentation, and differences between the analytical methods used. Indeed, our results show a hypovitaminosis D of 94.2% in women with breast cancer and 95.2% in women without breast cancer. The main mechanism of vitamin D deficiency results from the racial characteristic of being black, i.e., a cutaneous pigmentation. The melanin produced in the deep layers of the epidermis works as a filter that absorbs ultraviolet B radiation (UVB) in competition with 7-dehydrocholesterol [19]. Intense sunlight in sub-Saharan Africa may therefore compensate for low sun absorption through the skin [17]. In the current study, the higher concentration of 25(OH)D in the BC group could be explained by the significantly higher (*p* < 10^3^) sun exposure time for women with breast cancer (2.48 ± 0.62 h) compared with those without breast cancer (1.80 ± 0.80 h). Moreover, Chauveau et al. [19] have proven that UV doses must be multiplied by six in people of sub-Saharan African origin in order to obtain an identical concentration of vitamin D to that found in Caucasians. It is therefore worth noting that, in the current study, the mean exposure time in the cases and the control group was not sufficient, which probably led to 25(OH)D deficiency in comparison groups.

Analysing 1,25(OH)_2_D allows us to better understand 25(OH)D levels. A nonsignificant (*p* = 0.285) decrease in the second metabolite, 1,25(OH)_2_D, has been observed in women with breast cancer compared with women without breast cancer (Figure 1). Only a few studies have looked at the assay of both metabolites, 25(OH)D and 1,25(OH)_2_D, in parallel [18,24,28,29]. Compared with our outcomes, Lyra et al. [24] found a lower significant level of 1,25(OH)_2_D (*p* = 0.011) in the BC group. In keeping with our data, Janowsky et al. [18] found no significant fluctuation of 1,25(OH)_2_D between black women in their case and control groups, while there was a significantly lower concentration in white women with breast cancer compared with the controls. In the same study, there were no differences in mean value for either group with respect to 25(OH)D. Our results are in agreement with those of Hiatt et al. [30] and Bertone-Johnson et al. [28], with no difference in 1,25(OH)_2_D concentrations between cases and matched control subjects. The differences between all these studies and the current one confirm just how controversial the influence of plasma vitamin D on breast cancer is, showing that it could be explained using the same reasons cited above for 25(OH)D. In light of our results, there may be racial differences in the relationship between vitamin D and breast cancer. Indeed, our data agree with other findings regarding outcomes in black women, such as Janowsky et al.’s [18], while we observe the opposite with outcomes in white women [28,30]. The hypothesis of racial disparities is now common. Yao and Ambrosone [17] have affirmed that the high prevalence of vitamin D deficiency in women of African ancestry could be attributed to some degree to their ancestral genetic background, shaped over millennia in Africa. Differences in the timing of measurement could explain some of the variability in the study’s findings.

We performed a multivariate logistic regression model to better understand the involvement of vitamin D metabolites as factors linked to the risk of breast cancer. By stratifying by quartile levels, we found it noteworthy that there was a decrease in risk from the first quartile. These results show a possible association of 25(OH)D and 1,25(OH)_2_D in the decrease in breast cancer risk. Indeed, a number of studies have demonstrated 1,25(OH)_2_D’s antiproliferative, proapoptotic, prodifferentiating, and anti-inflammatory effects on malignant cells in breast cancer [7,31,32].

Our data indicate that 1,25(OH)_2_D remained in the normal range and has no level differences (Figure 1), despite mean values close to the lower limit of our technique showing a drop in concentration across the two groups. This decrease in levels of 1,25(OH)_2_D in the cases and controls suggests that its local production seems impaired and may be correlated with low levels of 25(OH)D detected in the same groups. This lends weight to the possibility that mechanisms in addition to hypovitaminosis D could be responsible for the association between breast cancer risk and the lower levels of 25(OH)D. Bertone Jonhson’s study [33] asserts that in case–control studies, the presence of a tumour may affect circulating vitamin D levels, either by altering 25(OH)D metabolism or by altering a patient’s dietary intake of vitamin D or sunlight exposure. Recent studies have also suggested that 25(OH)D is hydroxylated to form 1,25(OH)_2_D at extrarenal sites, including breast tissue [28]. The 1,25(OH)_2_D produced through this mechanism seems to act only as an autocrine or paracrine hormone, and it does not enter general circulation and may not be measurable by standard plasma assay [28,34]. In contrast, 25(OH)D levels are more sensitive to changes in diet and sunlight exposure and may not reflect the level of 1,25(OH)_2_D ultimately available to the target tissue [28,35].

The levels of vitamin D metabolites according to the stages of BC are necessary for a good understanding of the effects of metabolites. The majority of our breast cancer subjects were in advanced stages of the disease, with 83% for 1,25(OH)_2_D and 74.03% for 25(OH)D. We noticed no differences in plasma concentrations between early and advanced breast cancer levels of 25(OH)D and 1,25(OH)_2_D. There were also some similar concentrations of these metabolites between women with metastasis and without metastasis. These results could be explained by the small size of early breast cancer and women with metastasis. According to many studies, hypovitaminosis D in breast cancer could also be explained by the presence of VDR, because its expression in BC tissue decreases during tumour progression, making it less sensitive to vitamin D_3_ [36,37,38,39].

However, this study has some limitations, such as a lack of an evaluation of the enzymes 1α-hydroxylase, 24-hydroxylase, and the VDR in breast cancer tissue samples. This evaluation could provide more information about the mechanisms involved in the downregulation of 1,25(OH)_2_D. Another possible limitation is the low sample size of 1.25(OH)_2_D.

## 5. Conclusions

In the last decade, breast cancer has become the most frequently diagnosed cancer in Mali, and it continues to be a serious public health problem. Our data lend weight to the multifactorial aspects of this disease’s occurrence. In summary, our results prove that the population always consults late, which leads to a high rate of advanced breast cancer. The current study showed a possible association of 25(OH)D and 1,25(OH)_2_D in a decreased risk of breast cancer. Despite all the findings regarding the role of vitamin D in breast cancer disease, a number of questions remain to be answered, including the function of the vitamin D mechanism of dysregulation in breast cancer. Our study is the first to analyse 25(OH)D and 1,25(OH)_2_D in the blood of breast cancer patients in Mali. Given the level of disagreement in this field, further studies are needed to clarify the real impact of vitamin D metabolites in breast cancer disease.

## Figures and Tables

**Figure 1 diagnostics-13-03664-f001:**
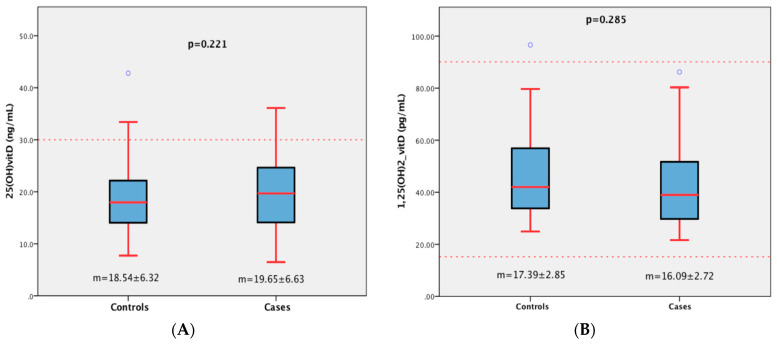
Plasma levels of 25(OH)D (**A**) and 1,25(OH)_2_D (**B**) in women with and without breast cancer. m: mean.

**Figure 2 diagnostics-13-03664-f002:**
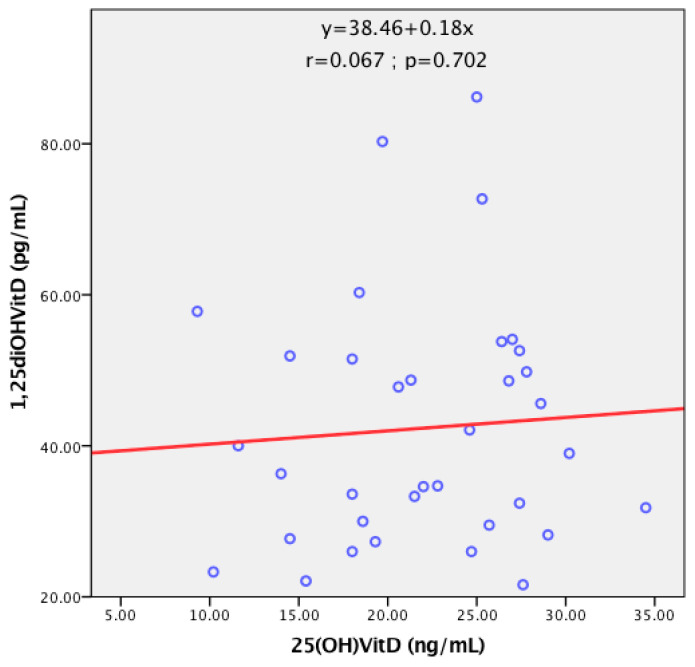
Scatterplot of the relationship between circulating levels of 25(OH)D and 1,25(OH)_2_D.

**Table 1 diagnostics-13-03664-t001:** Sociodemographic and clinico-pathological characteristics.

Characteristic		Cases (*n* = 104)	Controls (*n* = 104)	*p*-Value
Age (y *), mean ± SD **		48.09 ± 12.42	47.79 ± 12.42	0.863
Age at menarche (y), mean ± SD		14.32 ± 1.52	14.23 ± 1.35	0.715
First pregnancy age (y), mean ± SD		20.16 ± 4.12	22.56 ± 4.96	0.001
BMI *** (kg/m^2^), mean ± SD		26.39 ± 6.15	28.35 ± 6.21	0.024
Parity, mean ± SD		5.53 ± 2.93	3.01 ± 2.13	<10^−3^
Months of breastfeeding, mean ± SD		20.41 ± 8.76	15.92 ± 8.05	<10^−3^
Sleep time (hours)		7.17 ± 1.19	6.90 ± 1.61	0.172
Sun exposure time (hours)		2.48 ± 0.62	1.80 ± 0.80	<10^−3^
Residence	Urban, *n* (%)	67 (64.40)	99 (95.20)	<10^−3^
	Rural, *n* (%)	37 (35.60)	5 (4.80)	
Menopausal status	Menopausal women, *n* (%)	52 (50)	52 (50)	1
	Premenopausal women, *n* (%)	52 (50)	52 (50)	
Use of oral contraceptives	Yes, *n* (%)	29 (27.90)	51 (49)	0.002
	No, *n* (%)	75 (72.10)	53 (51)	
Menstrual cycle	Regular, *n* (%)	90 (86.50)	84 (80.80)	0.261
	Irregular, *n* (%)	14 (13.5)	20 (19.20)	
Smoking status	Current, *n* (%)	37 (35.60)	22 (21.20)	0.176
	Never, *n* (%)	67 (64.40)	82 (78.80)	
Professional status	Cleaning lady, *n* (%)	64 (61.53)	48 (46.15)	<10^−3^
	Entrepreneur, *n* (%)	25 (24.03)	28 (26.92)	
	State official, *n* (%)	9 (8.65)	11 (10.57)	
	Student, *n* (%)	0 (0)	2 (1.92)	
	Retiree, *n* (%)	4 (3.84)	11 (10.57)	
	Health official, *n* (%)	2 (1.92)	4 (3.84)	
Socioeconomic status ****	High, *n* (%)	10 (9.60)	24 (23.10)	<10^−3^
	Moderate, *n* (%)	21 (20.20)	60 (57.70)	
	Low, *n* (%)	73 (70.20)	20 (19.20)	
Education	Uneducated, *n* (%)	77 (74)	14 (13.50)	<10^−3^
	High school, *n* (%)	9 (8.70)	18 (17.30)	
	College graduate, *n* (%)	6 (5.80)	10 (9.60)	
	University, *n* (%)	12 (11.50)	62 (59.6)	
Family history of cancer	Yes, *n* (%)	13 (12.50)	23 (22.10)	0.049
	No, *n* (%)	91 (87.50)	81 (77.90)	
Past abortion	Yes, *n* (%)	47 (45.20)	52 (50)	0.277
	No, *n* (%)	57 (54.80)	52 (50)	
Dress habits *****	Loincloth and scarf set, *n* (%)	88 (84.51)	87 (83.65)	0.569
	Hijab set, *n* (%)	16 (15.39)	17 (16.35)	
Diet	Yes, *n* (%)	79 (75.90)	33 (31.70)	<10^−3^
	No, *n* (%)	25 (24.10)	71 (68.30)	
Dairy products	At least once a day, *n* (%)	55 (52.88)	61 (58.65)	0.248
	Sometimes, *n* (%)	12 (11.50)	14 (13.50)	
	Rarely, *n* (%)	37 (35.6)	25 (24)	
	Never, *n* (%)	0	4 (3.80)	
Meat consumption	At least once a day, *n* (%)	87 (83.65)	70 (67.30)	0.059
	Sometimes, *n* (%)	11 (10.60)	22 (21.20)	
	Rarely, *n* (%)	16 (15.80)	11 (10.60)	
	Never, *n* (%)	0	1 (1.00)	
Fish consumption	At least once a day, *n* (%)	84 (80.76)	81 (77.88)	0.892
	Sometimes, *n* (%)	17 (16.30)	18 (17.30)	
	Rarely, *n* (%)	3 (2.90)	5 (4.80)	
Fruit consumption	At least once a day, *n* (%)	86 (82.69)	81 (77.88)	0.044
	Sometimes, *n* (%)	12 (11.50)	16 (15.40)	
	Rarely, *n* (%)	6 (5.80)	7 (6.70)	
Physical activity	Yes, *n* (%)	100 (96.15)	104 (100)	0.043
	No, *n* (%)	4 (3.80)	0 (0.00)	
Tumour grade	T1, *n* (%)	2 (2.10)		
	T2, *n* (%)	11 (11.7)		
	T3, *n* (%)	17 (18.10)		
	T4, *n* (%)	64 (68.10)		
Nodal status	N0, *n* (%)	12 (12.80)		
	N1, *n* (%)	63 (67.00)		
	N2, *n* (%)	16 (17.00)		
	N3, *n* (%)	3 (2.60)		
Metastasis	M0, *n* (%)	69 (73.40)		
	M1, *n* (%)	25 (26.60)		
Clinical stage	I, *n* (%)	2 (2.10)		
	II, *n* (%)	15 (16.00)		
	III, *n* (%)	52 (55.30)		
	IV, *n* (%)	25 (26.60)		

* y: years; ** SD: standard deviation; *** BMI: body mass index; **** socioeconomic status: the criteria for socioeconomic status stratification was an estimation based on the answers to questions about income per month; ***** dress habits: most of our subjects were dressed as Malian women with a combination of loincloth and scarf sets. Loincloths are a complete cloth wrapped around the waist and with a top to match.

**Table 2 diagnostics-13-03664-t002:** Vitamin D levels and stage of advanced breast cancer.

		Study Cohort		*p*
		Cases	Controls	
Plasma 25(OH)D	Deficient, *n* (%)	54 (51.9)	65 (62.5)	0.303
	Insufficient, *n* (%)	44 (42.3)	34 (32.7)	
	Normal, *n* (%)	6 (5.8)	5 (4.8)	
Plasma 1,25(OH)_2_D	Normal, *n* (%)	35 (100)	34 (97.3)	0.327
	Low, *n* (%)	0 (0)	1 (2.7)	
	High, *n* (%)	0 (0)	0 (0)	

**Table 3 diagnostics-13-03664-t003:** Plasma levels of 25(OH)D (ng/mL) and 1,25(OH)_2_D (pg/mL) in case subjects classified by clinical stage and metastasis.

		25(OH)D	1,25(OH)_2_D
Clinical Stages I and II		(*n* = 17) 17.10 ± 6.50 ^1^	(*n* = 6) 34.95 ± 12.53 ^2^
Clinical Stages III and IV		(*n* = 77) 20.08 ± 6.57 ^1^	(*n* = 29) 43.84 ± 16.50 ^2^
Metastasis	No	(*n* = 69) 19.71 ± 6.77 ^3^	(*n* = 25) 42.99 ± 17.79 ^4^
	Yes	(*n* = 25) 18.89 ± 6.12 ^3^	(*n* = 10) 39.62 ± 11.60 ^4^

student *t* test; ^1^ 0.101; ^2^ 0.170; ^3^ 0.578; ^4^ 0.528.

**Table 4 diagnostics-13-03664-t004:** Multiple binary logistic regression model showing the association between quartile concentrations of 25(OH)D and 1,25(OH)_2_D in women in the case and control groups.

	*p*-Value	OR ^2^	95% CI ^3^	
			Lower	Upper
25(OH)D (*n* = 208) *	0.13	1.08	0.98	1.18
Q ^1^1 (<14.10 ng/mL)	0.25	0.40	0.08	1.91
Q2 (14.10–18.84 ng/mL)	0.16	0.32	0.06	1.60
Q3 (18.85–23.51 ng/mL)	0.23	0.38	0.08	1.80
Q4 (≥23.52 ng/mL)	1.00			
1,25(OH)_2_D (*n* = 70) **	0.18	0.97	0.92	1.02
Q1 (<31.92 pg/mL)	0.03	71.84	1.36	3785.34
Q2 (31.92–39.79 pg/mL)	0.98	0.96	0.07	14.28
Q3 (39.80–54.01 pg/mL)	0.98	0.96	0.05	19.50
Q4 (≥54.02 pg/mL)	1.00			

^1^ Q: quartile; ^2^ OR: odds ratio; ^3^ 95% CI: 95% confidence interval. * Logistic regression model, adjusted for menopausal status, age, age of menarche, body mass index, residence, smoking status, use of oral contraceptives, work, socioeconomic status, sleep time, education, family history of cancer, parity, first pregnancy age, months of breastfeeding, diet, dress habits, sun exposure time, and physical activity. ** Logistic regression model with restricted adjustment for menopausal status, age, age of menarche, body mass index, smoking status, use of oral contraceptives, socioeconomic status, sleep time, education, family history of cancer, parity, and months of breastfeeding.

## Data Availability

The data are not publicly available due to patient privacy regulations.

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
