# Peer review of "Plasma 25-Hydroxyvitamin D and 1,25-Dihydroxyvitamin D Levels in Breast Cancer Risk in Mali: A Case–Control Study"

_diagnostics, 2023, doi:10.3390/diagnostics13243664_

Round 1

Reviewer 1 Report

Comments and Suggestions for Authors

The manuscript entitled "Plasma 25-Hydroxyvitamin D and 1,25-Dihydroxyvitamin D lev- 2 els in breast cancer risk in Mali: a case-control study" by Bissan et al, well written and presented.

Authors please provide the Table 1 as landscape style (Some of the details are away from margin; not visible)

Author Response

Agreeing the Table 1, we made it in the landscape style in the new version of the manuscript.

We sincerely thank the Reviewer 1 for his/her relevant and constructive feedback which helped us to improve the manuscript. We hope that the present version will be found acceptable for publication.

Reviewer 2 Report

Comments and Suggestions for Authors

The aim of the research presented in this manuscript was to check whether there is a relationship between the risk of breast cancer and the amount of vitamin D3 in the blood in Malian women.
General comments:
1. The 25(OH)D level was tested in 110 women in each group (BC and control), while the 1,25(OH)2D level was tested in only 35 women from each group. What was the reason for limiting the number of women from the BC and control groups in whose the 1,25(OH)2D level was tested and on what basis was their selection made?
2. The description of the methodology for determining the 25(OH)D and the 1,25(OH)2D levels requires specification (what method was used, what kits or reagents were used, producers/suppliers of the necessary materials or reagents, antibody concentration and incubation time with them, etc.). With the current version of the methodology description in this area, it is not possible to repeat them.
3. Abbreviations referring to calcidiol [25(OH)D or 25(OH)D3] and calcitriol [1,25(OH)2D or 1,25(OH)2D3] to be standardized throughout the whole manuscript (text and figures).
4. I am unable to analyze the content in Table 1. It seems that the information on the right has been "cut off" (at least on pages 4 and 5 of 12, i.e. a problem with formatting the manuscript.
5. It would be easier for a potential reader to follow the information in the manuscript if two group names were consistently used: the control group (controls) and the breast cancer group (BC group).
6. It is a bit strange that no source of research funding was indicated.
Specific comments:
1. Line 3: no full stop
2. Lines 6 & 8: no country specified
3. Lines 10, 12, 14, 15, 16 & 18: no city specified
4. Lines 18: all words written with a capital letter
5. Line: 72: should be: 8 months, not: 8 months6. Line 74: all words in the name of the University written with a capital letter
7. Line 105: What do three dots mean? Shouldn't it be rather etc.
8. Line 112: Please specify rcf instead of rpm.

Comments on the Quality of English Language

minor editing

Author Response

We are sincerely grateful for your comments and suggestions concerning our article. Please find below our point-by-point responses. The new version of the manuscript has been modified accordingly.

Requested corrections are highlighted in yellow, while English corrections are in tracking word.

 Reviewer 2

The aim of the research presented in this manuscript was to check whether there is a relationship between the risk of breast cancer and the amount of vitamin D3 in the blood in Malian women.

General comments:
1. The 25(OH)D level was tested in 110 women in each group (BC and control), while the 1,25(OH)2D level was tested in only 35 women from each group. What was the reason for limiting the number of women from the BC and control groups in whose the 1,25(OH)2D level was tested and on what basis was their selection made?

Authors

The 1,25(OH)D was tested in only 35 women in each group because the decision to add this metabolite was taken after the 25(OH)D assay. Indeed, after all the analyzes of the plasma aliquots for 25(OH)D, there were only 35 aliquots left that could be used.

Reviewer 2

  1. The description of the methodology for determining the 25(OH)D and the 1,25(OH)2D levels requires specification (what method was used, what kits or reagents were used, producers/suppliers of the necessary materials or reagents, antibody concentration and incubation time with them, etc.). With the current version of the methodology description in this area, it is not possible to repeat them.

Authors

We are not agree with the reviewer on this point because the method described in the manuscript is an automatic method assay and we specified the method and the device as: chemiluminescence method on IDS-iSYS device. This method can be repeated only if the same device and the same reagents kits are used.

This point has been clarified for kits reagents in the revised version of the manuscript.

Reviewer 2

  1. Abbreviations referring to calcidiol [25(OH)D or 25(OH)D3] and calcitriol [1,25(OH)2D or 1,25(OH)2D3] to be standardized throughout the whole manuscript (text and figures).

Authors

All the abbreviations referring to calcidiol (25(OH)D) and calcitriol (1,25(OH)2D) are now standardized throughout the whole manuscript. This point has been clarified in the revised version of the manuscript.

Reviewer 2

  1. I am unable to analyze the content in Table 1. It seems that the information on the right has been "cut off" (at least on pages 4 and 5 of 12, i.e. a problem with formatting the manuscript.

Authors

We agreeing with the formatting problem of Table 1, we made it in the landscape style in the new version of the manuscript.

Reviewer 2

  1. It would be easier for a potential reader to follow the information in the manuscript if two group names were consistently used: the control group (controls) and the breast cancer group (BC group).

Authors

We agree and we have replaced two groups names by: control group (controls) and breast cancer group (BC group).

Reviewer 2

  1. It is a bit strange that no source of research funding was indicated.

Authors

This research is a phD one. The principal investigator is currently registered in Rabat, Morocco without funding for his research.

Specific comments:

Reviewer 2

1.Line 3: no full stop

Authors

We delete the full stop in line 3.

Reviewer 2

  1. Lines 6 & 8: no country specified

Authors

We specified the country : Morocco.

Reviewer 2

  1. Lines 10, 12, 14, 15, 16 & 18: no city specified

Authors

We specified city in the manuscript.

Reviewer 2

  1. Lines 18: all words written with a capital letter

Authors

We had written all words with a capital letter in line 18

Reviewer 2

  1. Line: 72: should be: 8 months, not: 8 months

Authors

We corrected the text in line 72: 8 months.

Reviewer 2

  1. Line 74: all words in the name of the University written with a capital letter

Authors

The name of the University has been written with a capital letter.

Reviewer 2

  1. Line 105: What do three dots mean? Shouldn't it be rather etc.

Authors

Indeed, we replaced three dots by “etc.”

Reviewer 2

  1. Line 112: Please specify rcf instead of rpm.

Authors

We specified rcf in line 112.

We sincerely thank the Reviewer 2 for his/her relevant and constructive feedback which helped us to improve the manuscript. We hope that the present version will be found acceptable for publication.

Sincerely yours,         

Aboubacar Dit Tietie Bissan

Reviewer 3 Report

Comments and Suggestions for Authors

Here, the authors presented a prospective case-control study analyzing a potential correlation between VitD serum levels and the development of breast cancer. Although there are many studies in this field, there is still lacking evidence, especially focusing on single populations and characteristics, such as race and ethnicity. Although this study could not reveal significant correlations, the work is appreciated and provides important information for the field.

However, there are several concerns, which have to be addressed before publication. Additionally, the overall presentation has to be extensively revised.

Major revision

Major concern is mainly about the use of 25(OH)D and 1,25(OH)2D3 as markers for VitD serum levels:
1. Why did the authors choose to measure 25(OH)D and 1,25(OH)2D3, considering that 25(OH)D is more stable and reliable for evaluating vitamin D status in the blood?
2. Lines 80-82: What are the selection criteria for the 35 women undergoing
1,25(OH)2D3 analysis? Why specifically 35, and why these women in particular?
3. Lines 209-210: The conclusions are based on 1,25(OH)2D, which is inappropriate as the data were only drawn from 35 patients without specific selection criteria. The fluctuation of 1,25(OH)2D makes it an unreliable marker for vitamin D status in the body.

Minor

4. Table 1 is clipped off and not readable
5. Line 50: The hydroxylation position is inaccurately identified as carbon 1α.
6. Line 61: The objective characterization on line 61 is too general. It is recommended to provide a more specific characterization of sunshine quality.
7. Line 63: The sentence is not accurate. Please verify the references.
8. Ethical approval was obtained in 2022, and sample collection started in August 2021.
9. Line 113: Please specify the duration samples were kept at -80°C.
10. Line 126: The sufficient level is 30-50 ng/ml, not only more than 30 ng/ml.
11. Table 1: The classification of the work could be based on the nature and time spent outdoors, not solely on job type.
12. Table 1: Specify the criteria for socioeconomic status stratification. Customized stratification may vary between countries, so accurate clarification is recommended.
13. Dress habits: Clarify dress habits, for example, what is meant by "Loincloth," etc.
14. Line 174: Clarify the significance of the presence or absence of the correlation between 25(OH)D and 1,25(OH)2D.
15. Table 2: The author used different nomenclature for the 25(OH)D classification (normal vs. previously indicated as sufficient).
16. Discussion (general comment): The authors referenced different research without explaining why their results differed from others.
17. The transition between different aspects in the discussion is not coherent and incomplete.
18. A professional scientific English revision is strongly recommended, eg Lines 310-311 and others.
19. Please revise the referencing, for example, line 302 [36–38] [39].

Comments on the Quality of English Language

English needs revision, grammar, spelling and punctuation should be revised by professional proofreading service.

Author Response

We are sincerely grateful for your comments and suggestions concerning our article. Please find below our point-by-point responses. The new version of the manuscript has been modified accordingly.

Requested corrections are highlighted in yellow, while English corrections are in tracking word.

Reviewer 3

Here, the authors presented a prospective case-control study analyzing a potential correlation between VitD serum levels and the development of breast cancer. Although there are many studies in this field, there is still lacking evidence, especially focusing on single populations and characteristics, such as race and ethnicity. Although this study could not reveal significant correlations, the work is appreciated and provides important information for the field.

However, there are several concerns, which have to be addressed before publication. Additionally, the overall presentation has to be extensively revised.

Major revision

Major concern is mainly about the use of 25(OH)D and 1,25(OH)2D3 as markers for VitD serum levels:

  1. Why did the authors choose to measure 25(OH)D and 1,25(OH)2D3, considering that 25(OH)D is more stable and reliable for evaluating vitamin D status in the blood?

Authors

We decided to measure the 1,25(OH)2D3 for two reasons. First, this metabolite is the biological active form of vitamin D3, which could help us to explain some of the variations of 25(OH)D levels. In fact, the lack of correlation between these two metabolites could make us hypothesize the presence of another unknown mechanism which could constitute new aims of research. Indeed, the current study shows the decrease of 1,25(OH)2D levels in cases and controls, suggesting that its local production seems impaired and maybe correlated to low levels of 25(OH)D.

The second reason was the automation possibility assay of the 1,25(OH)2D3, which helps avoid analytical errors.

Reviewer 3

2.Lines 80-82: What are the selection criteria for the 35 women undergoing
1,25(OH)2D3 analysis? Why specifically 35, and why these women in particular?

Authors

There were no selection criteria defined for the The 35 women undergoing
1,25(OH)2D3 analysis. In fact, 1,25(OH)D was tested in only 35 women in each group because the decision to add this metabolite was taken after the 25(OH)D assay. Indeed, after all the analyzes of the plasma aliquots for 25(OH)D, there were only 35 aliquots left that could be used.

Reviewer 3

  1. Lines 209-210: The conclusions are based on 1,25(OH)2D, which is inappropriate as the data were only drawn from 35 patients without specific selection criteria. The fluctuation of 1,25(OH)2D makes it an unreliable marker for vitamin D status in the body.

Authors

On this point we disagree with the Reviewer.

Despite us understand about the population size for 1,25(OH)2D assay, we consider that this is a preliminary study in our country and region, so these results could be a baseline for the future studies especially that we found a significant association considering 1,25(OH)2D.

Indeed, in this field, only a few studies have looked parallel to the assay of both metabolites, 25(OH)D and 1,25(OH)2D. For example, Lyra et al. [24] found a lower significant level of 1,25(OH)2D (p=0.011) in BC group with only 46 subjects and 31 for control group.

Minor

Reviewer 3

  1. Table 1 is clipped off and not readable

Authors

We agreeing with the formatting problem of Table 1, we made it in the landscape style in the new version of the manuscript.

Reviewer 3

  1. Line 50: The hydroxylation position is inaccurately identified as carbon 1α.

Authors

In this part we described the two hydroxylations steps that generate 1,25(OH)2D.

The first hydroxylation starts by carbon 25 by CYP2R1/CYP27A1 in the liver and the next in the kidney by CYP27B1 which hydroxylates 25-hydroxyvitamin D (25(OH)D) at carbon 1.

Reviewer 3

  1. Line 61: The objective characterization on line 61 is too general. It is recommended to provide a more specific characterization of sunshine quality.

Authors

We provided more specific characterization of sunshine quality. This part has been modified accordingly.

Reviewer 3

  1. Line 63: The sentence is not accurate. Please verify the references.

Authors

We modified the sentence and we verified the references too. The references are correct.

Reviewer 3

  1. Ethical approval was obtained in 2022, and sample collection started in August 2021.

Authors

The approval is obtained in 2021 but renewed in 2022. This 2022 number is an error.

We corrected the approval number in the manuscript.

Reviewer 3

  1. Line 113: Please specify the duration samples were kept at -80°C.

Authors

We specified the duration of conservation at -80°C.

Reviewer 3

  1. Line 126: The sufficient level is 30-50 ng/ml, not only more than 30 ng/ml.

Authors

We corrected the line 126 about the sufficient level.

Reviewer 3

  1. Table 1: The classification of the work could be based on the nature and time spent outdoors, not solely on job type.

Authors

We agree with reviewer 3 but we did not collect data on nature and time spent outdoors for the work’s classification.

Reviewer 3

  1. Table 1: Specify the criteria for socioeconomic status stratification. Customized stratification may vary between countries, so accurate clarification is recommended.

Authors

The criteria for socioeconomic status stratification was an estimation based on the answers of question about the income by month :

  • High status was those who had more than 1 million per month.
  • Moderate status were those with income between 350 000 and 1 million
  • Low status were defined by less than 350 000 by month

To avoid loading the table we have not specified these criteria. As in literature, these criteria were not specify.

Reviewer 3

  1. Dress habits: Clarify dress habits, for example, what is meant by "Loincloth," etc.

Authors

Most of our subject were dressed as Malians women with a combination of Loincloth and scarf set. Loincloth is a complete cloth wrapped around the waist and having a top to match.

Reviewer 3

  1. Line 174: Clarify the significance of the presence or absence of the correlation between 25(OH)D and 1,25(OH)2D.

Authors

We clarified the significance of the presence or absence of the correlation between 25(OH)D and 1,25(OH)2D in the manuscript.

Reviewer 3

  1. Table 2: The author used different nomenclature for the 25(OH)D classification (normal vs. previously indicated as sufficient).

Authors

We did not use different nomenclature about 25(OH)D classification. There is no sufficient status of 25(OH)D in the text.

Reviewer 3

  1. Discussion (general comment): The authors referenced different research without explaining why their results differed from others.

Authors

We agree with Reviewer 3 and we completed the discussion by the explanation of differences between our study and the others.

Reviewer 3

  1. The transition between different aspects in the discussion is not coherent and incomplete.

Authors

We added some sentences to make transition between discussion aspects, coherent.

Reviewer 3

  1. A professional scientific English revision is strongly recommended, eg Lines 310-311 and others.

Authors

A professional scientific English revised the manuscript.

Reviewer 3

  1. Please revise the referencing, for example, line 302 [36–38] [39].

Authors

The referencing has been revised especially line 302.

We sincerely thank the Reviewer 3 for his/her relevant and constructive feedback which helped us to improve the manuscript. We hope that the present version will be found acceptable for publication.

Sincerely yours,         

Aboubacar Dit Tietie Bissan

Round 2

Reviewer 1 Report

Comments and Suggestions for Authors

Accept 

Author Response

We are sincerely grateful for your comments and suggestions concerning our article.

Reviewer 2 Report

Comments and Suggestions for Authors

The authors responded to all my comments mentioned in my review and brought to the manuscript the necessary changes that would definitely enhance its quality.

Author Response

(The authors gave the same response as above.)

Reviewer 3 Report

Comments and Suggestions for Authors

Changes in the revised manuscript are appreciated, and authors indeed improved the manuscript.

However, as the raised major concerns could not be solved experimentally the authors must at least conclude their answers to the concerns in the final manuscript, especially the following:

- reasons or hypothesis why 25(OH)D and 1,25(OH)2D were chosen as markers for VitD serum levels (also mention the controversy or that the widely accpted marker is 25(OH)D)

- the fact that there are NO selection criteria for the 35 women/samples used for 1,25(OH)2D analysis MUST be in text. It is still written "we selected 35 women" which is NOT true! It was random!

- detailed information about cohort and definitions (see comments on dress habits, socioeconomic status etc) should be also published. If not included for example in footnotes of table 1, it can be also provided as supplementary information

Author Response

We are sincerely grateful for your comments and suggestions concerning our article. Please find below our point-by-point responses. The new version of the manuscript has been modified accordingly.

Requested corrections are highlighted in yellow, while English corrections are in tracking word.

Reviewer 3

Changes in the revised manuscript are appreciated, and authors indeed improved the manuscript.

However, as the raised major concerns could not be solved experimentally the authors must at least conclude their answers to the concerns in the final manuscript, especially the following:

  • reasons or hypothesis why 25(OH)D and 1,25(OH)2D were chosen as markers for VitD serum levels (also mention the controversy or that the widely accepted marker is 25(OH)D)

Authors

We added a short part to explain the reasons why we have chosen to measure 25(OH)D and 1,25(OH)2D as the first part of the discussion.

Reviewer 3

  • the fact that there are NO selection criteria for the 35 women/samples used for 1,25(OH)2D analysis MUST be in text. It is still written "we selected 35 women" which is NOT true! It was random!

Authors

We deleted the sentence with “selection mention” to new one which specify the random selection in the text.

Reviewer 3

  • detailed information about cohort and definitions (see comments on dress habits, socioeconomic status etc) should be also published. If not included for example in footnotes of table 1, it can be also provided as supplementary information

Authors

We resumed details and definition on dress habits and socioeconomic status in footnotes of table 1.

We sincerely thank the Reviewer 3 for his/her relevant and constructive feedback which helped us to improve the manuscript. We hope that the present version will be found acceptable for publication.

Sincerely yours,         

Aboubacar Dit Tietie Bissan